# FastClone is a probabilistic tool for deconvoluting tumor heterogeneity in bulk-sequencing samples

Yao Xiao[1,4], Xueqing Wang[1,4], Hongjiu Zhang[1,3,4], Peter J. Ulintz [2], Hongyang Li[1] & Yuanfang Guan [1,2 ✉]

Dissecting tumor heterogeneity is a key to understanding the complex mechanisms underlying drug resistance in cancers. The rich literature of pioneering studies on tumor heterogeneity analysis spurred a recent community-wide benchmark study that compares diverse modeling algorithms. Here we present FastClone, a top-performing algorithm in accuracy in this benchmark. FastClone improves over existing methods by allowing the deconvolution of subclones that have independent copy number variation events within the same chromosome regions. We characterize the behavior of FastClone in identifying subclones using stage III colon cancer primary tumor samples as well as simulated data. It achieves approximately 100-fold acceleration in computation for both simulated and patient data. The efficacy of FastClone will allow its application to large-scale data and clinical data, and facilitate personalized medicine in cancers.

[1] Department of Computational Medicine and Bioinformatics, Michigan Medicine, University of Michigan, Ann Arbor, MI, USA. [2] Department of Internal Medicine, University of Michigan, Ann Arbor, MI, USA. [3]Present address: Microsoft Inc., Redmond, WA, USA. [4]These authors contributed equally: Yao Xiao, Xueqing Wang, Hongjiu Zhang. ✉email: gyuanfan@umich.edu

Targeted therapy, a widely adopted cancer treatment[1], might have the potential to elicit promising initial responses, but many patients develop drug resistance during treatment[2,3]. Tumor heterogeneity is a major contributor to the development of drug resistance[1–4]. This heterogeneity is both spatial and temporal[3,5–7]. Spatial heterogeneity refers to the phenomenon that a tumor is composed of subclones of different genetic background[5,6], and temporal heterogeneity refers to the dynamic evolution of the tumor genomes through the disease course[7]. Therefore, when targeted therapy exerts its selection pressure, subclones with drug-resistant mutations would gradually dominate due to selective advantage[8]. Hence, dissecting the clonal composition of a tumor not only helps us to understand its biology and evolution but also guides the design of combinatorial therapies[2].

One strategy to deconvolute spatial and temporal tumor heterogeneity is to perform multiregional and longitudinal sampling[2]. However, this strategy may not be suitable under all circumstances because of the ethics of carrying out unnecessary invasive procedures as well as the practicality of longitudinal sampling for solid tumors, since the majority of cancer samples are obtained from surgical procedures, and if a tumor has been removed at the one-time point, it will not be available for sampling at a later time point[3]. In recent years, strategies have been developed by combining deep whole-genome sequencing (WGS) or whole-exome sequencing (WES) with novel statistical and computational methods. The workflow generally consists of two parts. First, bulk sequencing provides mixed information about genetic alterations for all subclones in the tumor, including allele frequency for single-nucleotide variations (SNVs) and copy number alteration (CNA). Then, machine learning models are used to cluster the SNVs into subclones and reconstruct the tumor phylogenetic trees[9]. A number of pioneering models based on the SNV and CNA profiles have been developed since 2013[10–27]. While applauding for these pioneer works including but not limited to PyClone, PhyloWGS, and SciClone, the field needed a standard for evaluating these algorithms[28]. This need was addressed by the DREAM Somatic Mutation Calling-Heterogeneity Challenge (DREAM SMC-Het Challenge), which evaluated the models on three aspects[28]: (1) evaluating each model's ability in predicting global traits, including tumor purity, the number of subclones, and the proportion of each subclone; (2) assessing each model's ability to assign SNVs to each subclone; (3) assessing each model's accuracy in inferring phylogenetic relationships[28]. This carefully designed evaluation scheme provides a solid benchmark for tumor heterogeneity modeling algorithms.

Here, we present FastClone, a probabilistic model for inferring tumor heterogeneity, which ranks as a top algorithm in the DREAM SMC-Het Challenge. In addition, FastClone excels in inference speed: it takes only a few seconds for inferring a tumor with tens of thousands of mutations. In this study, FastClone is applied to both computationally simulated data and deep sequencing stage III colon tumor samples acquired at Michigan Medicine. The efficiency and accuracy of the algorithm will allow FastClone to be applied in clinical research and very large-scale datasets.

## Results

### Overview of FastClone algorithm.
FastClone uses the bulk DNA-sequencing data of a single tumor sample as the input (Fig. 1). Information includes the copy number profile and allele frequencies of SNVs. FastClone starts by inferring the prevalence of cells that contain a certain SNV in the tumor sample ($\rho$). For each SNV, FastClone calculates $\rho$ based on the observed allele frequencies ($\beta$), the major and the minor copy numbers of CNA (denoted as $N_{major}$ and $N_{minor}$, respectively). In this regard, FastClone shares some similarities with previous work, such as PyClone[11], SciClone[15], and PhyloWGS[17]. The major differences between FastClone and other algorithms lie in the following two aspects: (1) greatly accelerating the inference process, and (2) improving the generalizability of the method to the scenario where CNV happened independently on the same section of chromosomes in different subclones (denoted as "2-state" or "multistate" below). This was achieved by first identifying subclones based on SNVs on nonambiguous chromosome sections, and then assigning all SNVs, regardless of the CNA status, to the subclones by maximal likelihood. Lastly, the phylogeny tree of the tumor is inferred by exhaustively listing out all possibilities and selecting the structure with the highest likelihood (Fig. 1). Supplementary Table 1 summarizes all the symbols used in the following equations.

**Estimating the prevalence of cells with a specific SNV.** The prevalence of cells that contain a specific SNV ($\rho$) can be directly calculated from the allele frequency of the SNV ($\beta$). However, the existence of CNA events complicates the situation. Thus, the calculation of $\rho$ is discussed under two situations—with or without CNA events. Sex chromosomes of males, since they only have one X chromosome, are discussed separately in both situations (Fig. 2).

When there is no CNA event (Fig. 2a-(1), b-(1)), the mutated allele would randomly appear in either one of the pair of autosomes or the sex chromosomes of a female (Fig. 2a-(1)). We denote the total cell number by $n_{cell}$. Then, the number of mutated alleles can be calculated as $n_{cell} \times \rho \times 1$, and the total number of alleles is $2n_{cell}$. Therefore, $\beta$ equals the proportion of mutated alleles in all alleles (Eq. (1)). To calculate $\rho$ from $\beta$, we inverted the function (Eq. (2))

$$\beta = n\,\text{cell} \times \rho \times 1/2\,n\,\text{cell} = \rho/2, \qquad (1)$$

$$\rho = 2\beta. \qquad (2)$$

For male's sex chromosomes (Fig. 2b-(1)), the allele frequency directly equals the proportion of tumor cells (Eqs. (3) and (4))

$$\beta = n\,\text{cell} \times \rho \times 1/n\,\text{cell} = \rho, \qquad (3)$$

$$\rho = \beta. \qquad (4)$$

In the below calculation, we will not repeat the process of division of $n_{cell}$ in both denominator and numerator.

When there are CNA events, they can happen only in one subclone (represented by 1-state below), or happen in both one parent and its child subclones, or happen in two independent subclones (represented by 2-state below). Along this line, CNA events could be more than 2-state. In the CNA data, we have a list of chromosome sections and their CNA values in each state. The relationship between the prevalence of cells that contain an SNV to the observed frequency is highly affected by CNA events. Our approach estimates the prevalence of the cells based on SNVs with no CNA or with 1-state CNA events to identify the number of subclones. We do not know the relative ratios of the subclones if the chromosome segment were 2-state or multistate and thus unable to infer the $\rho$ associated with the SNV event. Then, we assign each SNV to a subclone no matter which CNA states they are associated with by maximal likelihood.

CNA events can happen multiple times to the chromosome loci, and the order of occurrence of SNV and CNA events is not fixed. 1-State CNA events can thus be further divided into several situations. For these situations that the order of SNVs and CNA events are mixed together, we also consider whether the SNV is located on the

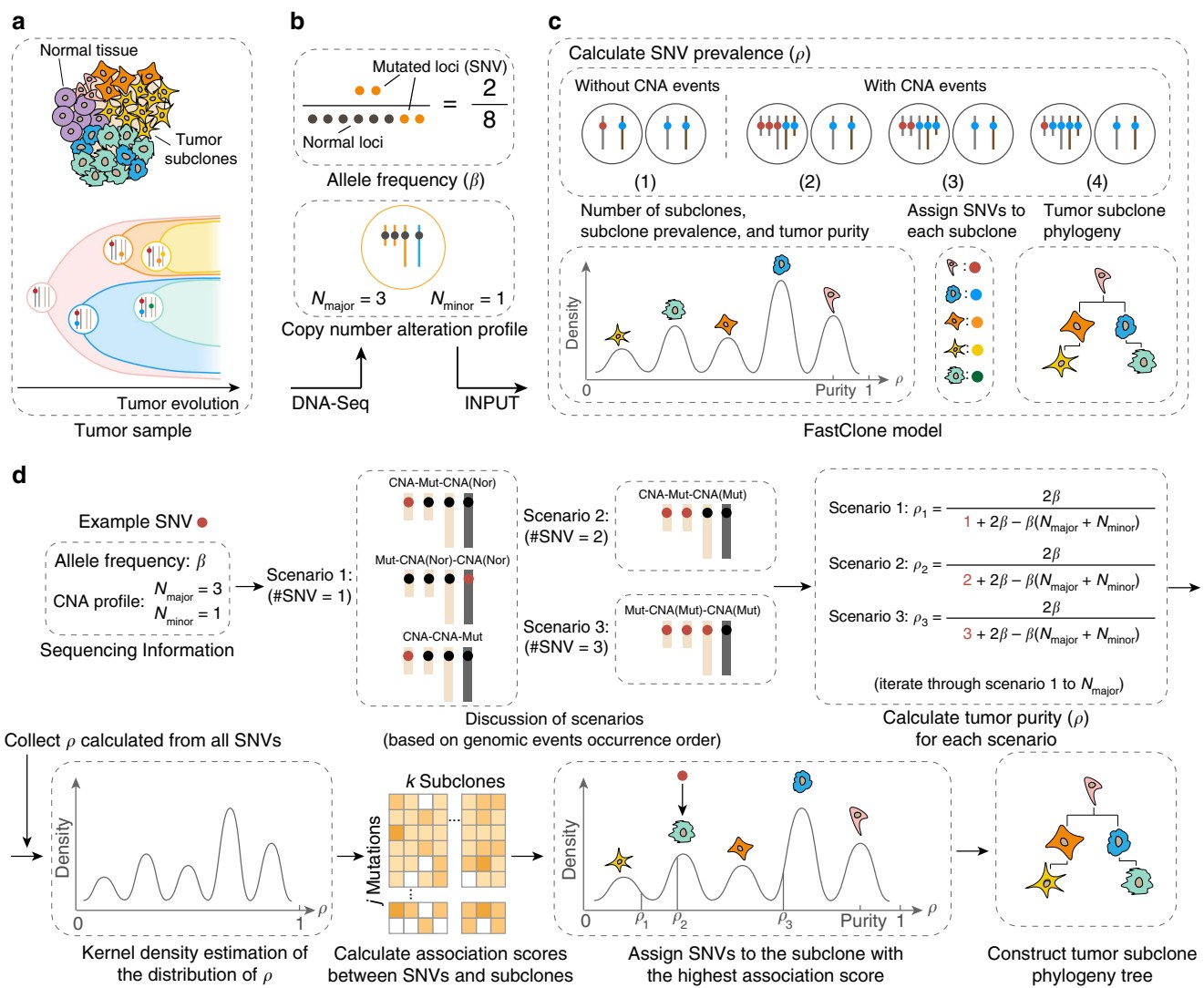

**Fig. 1 Overview of the FastClone algorithm. a** The tumor sample is heterogeneous, composed of both normal cells and tumor subclones (top panel). The tumor dynamically evolves throughout the disease course, generating subclones with different genotypes (bottom panel). The dots in different colors represent different SNVs. **b** DNA-sequencing of the bulk tumor provides information about (1) allele frequency of each SNV ($\beta$), that is, the observed allele occurrence among all cells (top panel); (2) a CNA profile in the form of $N_{major}$ and $N_{minor}$ (bottom panel). Each of the yellow dots represents a copy of the allele with a certain SNV. **c** FastClone model. First, we calculated the proportion of cells that carry each SNV ($\rho_i$ for SNV$_i$). This calculation is discussed in two situations: with and without CNA events. Multiple possibilities are further discussed (see Fig. 2). Blue spheres represent normal loci, and red spheres represent mutated loci that contain SNVs. Then, subclone numbers, subclone proportions, and tumor purity are determined from the distribution of $\rho$. After that, SNVs are assigned to subclones. Finally, the putative evolutionary relationship of the subclones is established. **d** The workflow of FastClone algorithm. The workflow starts with sequencing information as input, which includes allele frequency ($\beta$) and CNA profile ($N_{major}$ and $N_{minor}$). Since we do not know the order of occurrence of genomic events, all possible scenarios are discussed, and $\rho$ value for each scenario is calculated. The number of possible scenarios equals the value of $N_{major}$. Then, KDE is used to determine the distribution of $\rho$. Each peak in the $\rho$ distribution indicates a subclone. After that, association scores between each SNV–subclone pair are calculated. Then, the SNV is assigned to the subclone with the highest association score. If there are several $\rho$ values associated with one SNV, then the $\rho$ that provides the highest association score is used to assign the SNV to the subclone (in this case, $\rho_2$ that assigns this SNV to the green subclone is considered the correct solution). Finally, the most likely phylogeny tree of the subclones is constructed.

major copy (Fig. 2a-(2)) or the minor copy number (Fig. 2a-(3)). As one cannot directly determine the order of occurrence of genomic events and decide which set of equations to use, FastClone calculates the $\rho$ value of every possible scenario. Then, the association scores between SNVs and subclones for all scenarios are calculated separately, and the SNVs are assigned to the subclone with the highest association score among all scenarios. This association score will be different for each scenario, because if the scenario is correct, the $\rho$ will be more likely to be assigned to that peak (Fig. 1d). During the estimation of the number of clones, if there are enough SNVs on the chromosome sections without CNV (>100), we use these SNVs to estimate the number of clones. Otherwise, all scenarios get votes in the estimation of the number of subclones, because we do not know the exact ordering of SNV and CNV events. Although individually some of the scenarios are wrong, statistically across many SNVs, this can help us to generate the correct number of clones, especially together with the SNVs on chromosomes without CNAs. The steps of estimating subclone numbers and assigning SNVs to subclones would be described in detail in the next two sections.

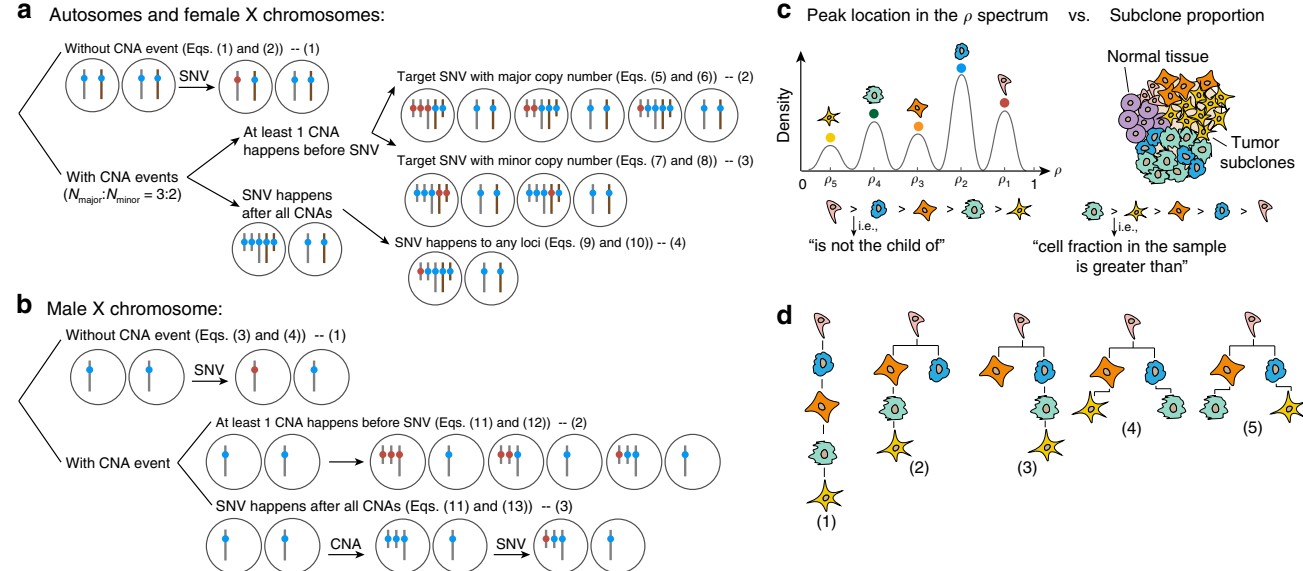

**Fig. 2 Calculating the prevalence of cells with target SNV ($\rho$) and predicting phylogenetic trees. a, b** Gray bars represent maternal chromosomes and brown bars represent paternal chromosomes, blue spheres represent normal loci, and red spheres represent mutated loci that contain SNVs. **c** The distribution of peaks' location on the $\rho$ spectrum (left panel) will allow the identification of subclones (right panel). Dots in different colors represent the SNVs that are assigned to the peaks. Intuitively, an SNV being assigned to a peak indicates this SNV occurs in the subclone that is related to this peak. The $i$th peak's location $\rho_i$ indicates that $\rho_i$ percent of the cells contain the SNVs assigned to this peak. Since all child subclones contain the SNVs inherited from their parent subclones, the $\rho$ value of child subclones should be smaller than their parent subclones. Therefore, a subclone with a smaller peak location in the $\rho$ spectrum cannot be the parent of a subclone associated with a peak of larger $\rho$. However, each subclone's fraction in the sample is not strictly related to the subclone's position in the phylogenetic tree. **d** Some examples of possible phylogenetic trees. Based on the hierarchy determined in **c**, several tree structures may be possible since all of them satisfy the above rules.

When the CNA event happens to the chromosome section that is already with an SNV (Fig. 2a-(2)), the SNV will be duplicated. The relationship between $\beta$ and $\rho$ can be calculated in a similar way as in Eqs. (1) and (5). Depending on the ordering relationship between the CNA events and the SNV, the equation for $\rho$ can be obtained by taking the inverse of Eqs. (5) and (6):

$$\beta = \frac{[1, 2, ..., N_{major}]\rho}{(N_{major} + N_{minor})\rho + 2(1 - \rho)}, \tag{5}$$

$$\rho = \frac{2\beta}{[1, 2, ..., N_{major}] + 2\beta - \beta(N_{major} + N_{minor})}. \tag{6}$$

When the target allele has fewer copies than the other allele (Fig. 2a-(3)), then in Eq. (5), the $\rho$ at the numerator position should be multiplied by the minor copy number instead of the major copy number (Eq. (7)). We inverted Eq. (7) to obtain the formula of $\rho$ (Eq. (8)):

$$\beta = \frac{[1, 2, ..., N_{minor}]\rho}{(N_{major} + N_{minor})\rho + 2(1 - \rho)}, \tag{7}$$

$$\rho = \frac{2\beta}{[1, 2, ..., N_{minor}] + 2\beta - \beta(N_{major} + N_{minor})}. \tag{8}$$

Of note, $N_{minor}$ can also be more than one. $N_{minor}$ can also be zero, that is, deletion, in which case, the conditions of Eqs. (7) and (8) are not considered. For the situation where the CNA event happens before an SNV (Fig. 2a-(4–1,2)), it does not matter whether the target allele has a major copy number (Fig. 2a-(4–1)) or a minor copy number (Fig. 2a-(4–2))—the resulting frequency would be the same in these two situations. Here, the mutation would only happen to one allele, so the $\rho$ in the numerator position of Eq. (5) should be multiplied by the constant value one,

which is simplified in the formula (Eqs. (9) and (10)):

$$\beta = \frac{\rho}{(N_{major} + N_{minor})\rho + 2(1 - \rho)}, \tag{9}$$

$$\rho = \frac{2\beta}{1 + 2\beta - \beta(N_{major} + N_{minor})}. \tag{10}$$

Finally, for the sex chromosomes of males (Fig. 2b), we removed $N_{minor}$ variant alleles and half of the normal alleles from the denominator of Eqs. (5) and (11). Then, we again calculated $\rho$ in two situations—the mutation occurs before a CNA event (Fig. 2b-(2), Eq. (12)) or after a CNA event (Fig. 2b-(3), Eq. (13))

$$\beta = \frac{[1, 2, ..., N_{major}]\rho}{N_{major}\rho + (1 - \rho)}, \tag{11}$$

$$\rho = \frac{\beta}{[1, 2, ..., N_{major}] + \beta - \beta \times N_{major}}, \tag{12}$$

$$\rho = \frac{\beta}{1 + \beta - \beta \times N_{major}}. \tag{13}$$

The above analysis is a comprehensive analysis of different scenarios. It is possible that an inferred $\rho$ is >1, when the scenario is an incorrect one. In such cases, the inferred value is discarded as it is impossible to have a prevalence bigger than one.

**Estimating the number of subclones and tumor purity.** We then identified the number of subclones based on all inferred prevalence values. First, we used Gaussian kernel density estimation (KDE) to reconstruct the distribution of allele frequency. From the reconstructed distribution, we extract the initial guess of the clusters by identifying local maxima in the distribution, which will be refined by more accurate probabilistic distributions.

Although this is a rough approximation, assuming Gaussian results in substantial acceleration in computation compared to alternatives such as beta distribution. This roughness is alleviated by the assignment of the SNVs later, which is modeled with the binomial distributions. Additionally, beta distribution tends to overestimate the number of clones, and tends to identify subclones where there is no separate peak[15]. For example, it is possible to generate clones with only one or two variants, which takes sophisticated methods to prune[15] and potentially leads to overestimation of the number of clones. While in our KDE estimation, we utilized Scott's rule to derive the number of clones and controls such overestimation.

Below, we show the KDE function of the distribution of $\rho$, where $n$ is the total number of SNVs, $\beta_j$ is the allele frequency of the $j$th SNV, $Z$ is the normalization constant to ensure $\int_{-\infty}^{+\infty} \hat{f}(\beta)d\beta = 1$, and $h$ is the bandwidth for smoothing density estimations to make it as close as possible to the true density (Eq. (14)). Here, we applied Scott's rule[29] to estimate the bandwidth $h$ with $d$-dimensional data ($d$ indicates the number of samples) (Eq. (15))

$$\hat{f}(\beta) = \frac{1}{nZ} \sum_{j=1}^{n} e^{\left(\frac{\beta - \beta_j}{h}\right)^2}, \tag{14}$$

$$h = n^{-\frac{1}{d+4}}. \tag{15}$$

FastClone then enumerates all local maxima of the density function $\hat{f}$. It creates a grid over the coordinate space with a resolution of 0.001, picks up any local maxima, and further optimizes the maxima locations through Nelder–Mead simplex algorithm. FastClone uses these local maxima to specify individual subclones.

Furthermore, based on the fact that all tumor cells inherit the mutations from their previous generations, there should be at least one SNV that exists in all tumor cells. Therefore, we use the biggest local maximum-associated $\rho$ value as the tumor purity.

**Assigning SNVs to subclones**. To further match the SNVs with each subclone, we calculated subclones' weights, which reflect the prevalence of the subclones in the tumor sample and the association score between each mutation and each subclone. The rationale behind calculating the weight of the subclone can be explained by a simple scenario where the prevalence of SNV lies in exactly the middle of two local maxima. Then, this SNV should be assigned to the cluster whose prevalence is higher.

The weight and the scores are estimated by optimizing the log-likelihood function (Eq. (16)). In Eq. (16), $C$ is the total number of peaks or subclones, $w_j$ is the proportion of the $j$th subclone, and $L_{jk}$ is the probability of the $k$th mutation associated with the $j$th subclone. Note that $w_j$ is different from $\rho_j$, which is defined as the proportion of cells that contain the $j$th SNV, because a subclone may contain several SNVs,

$$\lambda = \sum_{j=1}^{C} \ln \sum_{k=1}^{n} w_j L_{jk}. \tag{16}$$

For each mutation, its probability of being associated with subclones is modeled by several binomial distributions (Eq. (17)),

$$L_{jk} = \text{Binom}(m_k, r_k; \hat{\beta}_{jk}). \tag{17}$$

Here, $L_{jk}$ is the probability of the $k$th mutation associated with the $j$th subclone, $m_k$ is the observed reads that carry the $k$th mutation, $\hat{\beta}_{jk}$ is the expected allele frequency of the $k$th mutation if it is associated with the $j$th subclone, and $r_k$ is the total number of reads that cover the locus of the $k$th mutation and pass the

quality filter. Given the proportion $w_j$ of the $j$th subclone, the expected allele frequency of a mutation ($\hat{\beta}_{jk}$) can fall into the following two cases: SNV located on autosomes or female X-chromosomes or the SNV located on the male sex chromosomes. In addition, as we do not know the exact number of alleles that contain the SNV, we iterate through all possible situations and choose the one that results in the highest likelihood $L_{jk}$.

If the SNV is located on an autosome or female X-chromosome, there are three possibilities:

(1) If the mutation occurs on the major copy either before or in the middle of a sequence of CNA event, then,

$$\hat{\beta}_{jk,\text{major}} = \frac{[2, ..., N_{\text{major}}]\rho_j}{(N_{\text{major}} + N_{\text{minor}})\rho_j + 2(1 - \rho_j)}. \tag{18}$$

(2) If the mutation occurs on the minor copy before the CNA event, in several potential cases where there is or there is an absence of CNA on this minor copy, then,

$$\hat{\beta}_{jk,\text{minor}} = \frac{[1, 2, ..., N_{\text{minor}}]\rho_j}{(N_{\text{major}} + N_{\text{minor}})\rho_j + 2(1 - \rho_j)}. \tag{19}$$

If $N_{\text{minor}}$ is zero (deletion), then this situation is not considered, because SNV will no longer exist.

(3) If the mutation occurs on either copy after the CNA event,

$$\hat{\beta}_{jk,\text{after}} = \frac{\rho_j}{(N_{\text{major}} + N_{\text{minor}})\rho_j + 2(1 - \rho_j)}. \tag{20}$$

Then, $\hat{\beta}_{jk}$ will be the most likely value among the above cases (some of them are overlapping),

$$\hat{\beta}_{jk} = \text{argmax}_{\hat{\beta} \in \{\hat{\beta}_{jk,\text{major}}, \hat{\beta}_{jk,\text{minor}}, \hat{\beta}_{jk,\text{after}}\}} \text{Binom}(m_k, r_k; \hat{\beta}). \tag{21}$$

If the SNV is located in a male sex chromosome region, the case will be similar to the calculation above but with $N_{\text{minor}} = 0$. There is no mutation occurring on the minor copy, and thus $\hat{\beta}_{jk}$ will be the most likely value between (1) if the mutation takes place after copy number variation:

$$\hat{\beta}_{jk,\text{major}} = \frac{\rho_j}{N_{\text{major}} \times \rho_j + (1 - \rho_j)} \tag{22}$$

or (2) the mutation takes place before copy number variation:

$$\hat{\beta}_{jk,\text{major}} = \frac{[1, 2, ..., N_{\text{minor}}]\rho_j}{N_{\text{major}} \times \rho_j + (1 - \rho_j)}. \tag{23}$$

Then, $\hat{\beta}_{jk}$ will be the most likely value among these two:

$$\hat{\beta}_{jk} = \text{argmax}_{\hat{\beta} \in \{\hat{\beta}_{jk,\text{major}}, \hat{\beta}_{jk,\text{after}}\}} \text{Binom}(m_k, r_k; \hat{\beta}). \tag{24}$$

**Reconstructing the phylogeny tree**. The phylogeny construction is done by iterating through all possible tree structures and excluding the structures that are not possible. To this end, we must first differentiate two concepts: subclone proportion and its associated peak location in the $\rho$ spectrum (Fig. 2c). While the proportion of a parent subclone could be bigger or smaller than its children, the parent–child relationship for the associated peak locations must follow this rule: a subclone with a smaller peak location in the $\rho$ spectrum cannot be the parent of a subclone with a larger peak location.

Up until this point, we only obtained the number of subclones and their associated peak locations, as well as their associated SNPs. We do not yet know the tree structure (see example in Fig. 2d). Thus, in order to establish a valid tree, we enumerated all possible structures by examining the peak spectrum. Then, the associated proportion of each subclone is assigned. With all the possible tree topologies enumerated, FastClone attempts to rank

the topologies based on the predefined beta distribution prior that was explained above, assuming the clones containing more SNVs will be at lower (towards the child node direction) in the tree hierarchy. This assumption has its biological reason that as cancer progresses, the mutation rate is likely to be accelerated due to accumulated mutations in key genome stability pathways, such as DNA repair and replication pathways or cell-cycle checkpoints[30,31]. However, without other information, often multiple candidate topologies are valid. This selection was done with brute-force search of tree structure and comparison.

**FastClone achieved the highest accuracy on simulated data**. To test the performance of FastClone, we first applied it to the simulated benchmark data provided by the SMC Tumor Heterogeneity Challenge. Compared to tumor subclone data from databases and single-cell sequencing data, simulated benchmark data are a complementary approach to provide a gold standard for the reconstruction of subclones[28]. The SMC Tumor Heterogeneity Challenge generated the simulated data based on the BAMSurgeon tool[32,33], which generates synthetic tumors by adding mutations to existing reads, and added phasing of mutations, large-scale CNA, translocations, trinucleotide SNV signatures, and replication timing effects[28]. This design ensures that not only SNVs are created but complicated features in the real-world data are also represented in the simulated data[28]. We used eight simulated samples (Tumors 1, 2, 3, 4, 6, 7, A, and B) to test the accuracy of FastClone, ranging from 1 to 5 subclones. These simulated samples were based on the already-sequenced tumor cell line, which is sequenced by SAMtools[34], and their chromosome copy number variations were obtained by Nexus Copy Number software v7.5. The performance of the model was evaluated from three aspects: (1) the ability to predict overall properties of the bulk tumor, including tumor purity, number of subclones, and the cellular proportion of each subclone; (2) the ability to accurately match SNVs with subclones; (3) the ability to determine the phylogenetic order of the subclones.

Below, we will present the performance of the model through each sub-challenge. First, we present the best performances we get from the model that we further improved after the challenge. These scores are calculated using all the eight simulated tumor samples and the evaluation code provided by the challenge. Then, we compared the performance of FastClone against other programs.

Sub-challenge 1A evaluates accuracy in predicting tumor purity. Intuitively, it measures the absolute difference between the true and the estimated purity. Thus, a perfect prediction will achieve a score of one, and a random prediction will result in zero. The median score FastClone achieved on eight samples was 0.988 (Fig. 3a-1A). On the challenge leaderboard, we procured the same best result with another algorithm with a median performance score of 0.99, which is an excellent performance for simulated tumor samples (Supplementary Fig. 1). The performance scores of all other four software range from 0.37 to 0.89, with a median score of 0.77 (Fig. 3b)[35]. Sub-challenge 1B evaluates the accuracy in predicting the number of subclones, the performance is reflected by a score that is calculated as one minus the absolute difference between the true and the estimated number of subclones, divided by the maximal value of the prediction and the gold standard, so a score of one indicates a perfect prediction. The median score FastClone achieved on eight samples was 1 (Fig. 3a-1B). On the leaderboard, FastClone ranked second place among all software with a median score of 0.75 (Supplementary Fig. 1), and the scores of the other five software range from 0.38 to 1, with a median score of 0.67 (Fig. 3b)[35].

1C evaluates the accuracy in predicting the proportion for each subclone. The score is calculated by one minus the mean absolute difference between the true and the estimated distribution of cellular prevalence. In this part, a median score of 0.974 shows that the prediction we gave was very close to a perfect result (Fig. 3a-1C). On the leaderboard, FastClone obtained a score of 0.97 (Supplementary Fig. 1). The scores of the other five software range from 0.6 to 0.89, with a median score of 0.74 (Fig. 3b)[35]. These results suggest a robust performance of the model across a wide range of scenarios.

Sub-challenge 2 evaluates the performance of the algorithm on determining mutation assignments to subclones. The scoring metric is a mean of the correlations between the true and estimated co-clustering matrix of pairwise SNVs that are calculated by two different measures, and here, a random correlation is zero. Two different correlation measures were applied: first for the binary assignment accuracy and the second one for the adjacency matrix (i.e., probabilistic assignment where each SNV is associated with each subclone with a probability). We obtained median scores of 0.662 and 0.738 on eight tumor samples, respectively (Fig. 3a-2A, 2B). The leaderboard median scores of 0.47 and 0.6 place FastClone at the top in this sub-challenge (Supplementary Fig. 1)[35], and the scores of the other five software (for both sub-challenge 2A and 2B) range from 0.09 to 0.47, with a median score of 0.21 (Fig. 3b).

Sub-challenge 3 focuses on evaluating the prediction accuracy of subclone phylogeny, and the result is measured by a score that reflects the mean of the symmetric pseudo-V-measure correlation between the true and predicted ancestor-descendant matrix. For sub-challenges 3A and 3B, we achieved median scores of 0.851 and 0.875 on eight simulated tumors (Fig. 3a-3A, 3B). For sub-challenge 3A, we obtained a median score of 0.69, being the only team with a final submission on the leaderboard. There are no entries on the leaderboard for sub-challenge 3B.

Overall, FastClone has an excellent performance in predicting the tumor purity, the number of subclones, and the proportion of each subclone. In almost all sections, FastClone had the highest median score among all the models participated (Supplementary Fig. 1 and Fig. 3b)[35], and by visualizing the comparison between the ground truth and FastClone's prediction on one of the simulated samples, we can more clearly see this outstanding performance as the predictions are almost the same as the ground truths across all the sub-challenges (Fig. 3c). This outstanding performance in SMC-Het Challenge suggests that FastClone is a state-of-the-art model in the field of tumor subclone reconstruction.

To investigate whether the number of SNVs would affect FastClone estimation, we randomly subsampled the SNVs of the eight simulated tumor samples, and observed the changes of the predictions for purity and number of clones. For each tumor, we randomly sampled SNVs for 99 times: from 1 to 9% of the original number of SNVs. Overall, we found errors are rare for most tumors even with only dozens of SNVs (Supplementary Fig. 2). For the two tumors (Tumor 7 and Tumor B), which we did find a change in the predicted number of clones, the clones that were dropped off are the ones containing the least SNVs, for which the sampling was not sufficient. For example, Tumor 7, with a total of 2834 SNVs, had a clone of only 89 SNVs (3% of the total population). When we subsampled it to 200 SNVS, we had only about nine SNVs left, and this number was not statistically strong enough to support a separate clone. In conclusion, FastClone's predictions are mostly robust across a wide range of SNVs in tumor samples. Furthermore, we assessed how the number of subclones and the number of SNVs affect FastClone's accuracy of estimating the number of subclones through simulation experiments. As expected, the accuracy drops as the

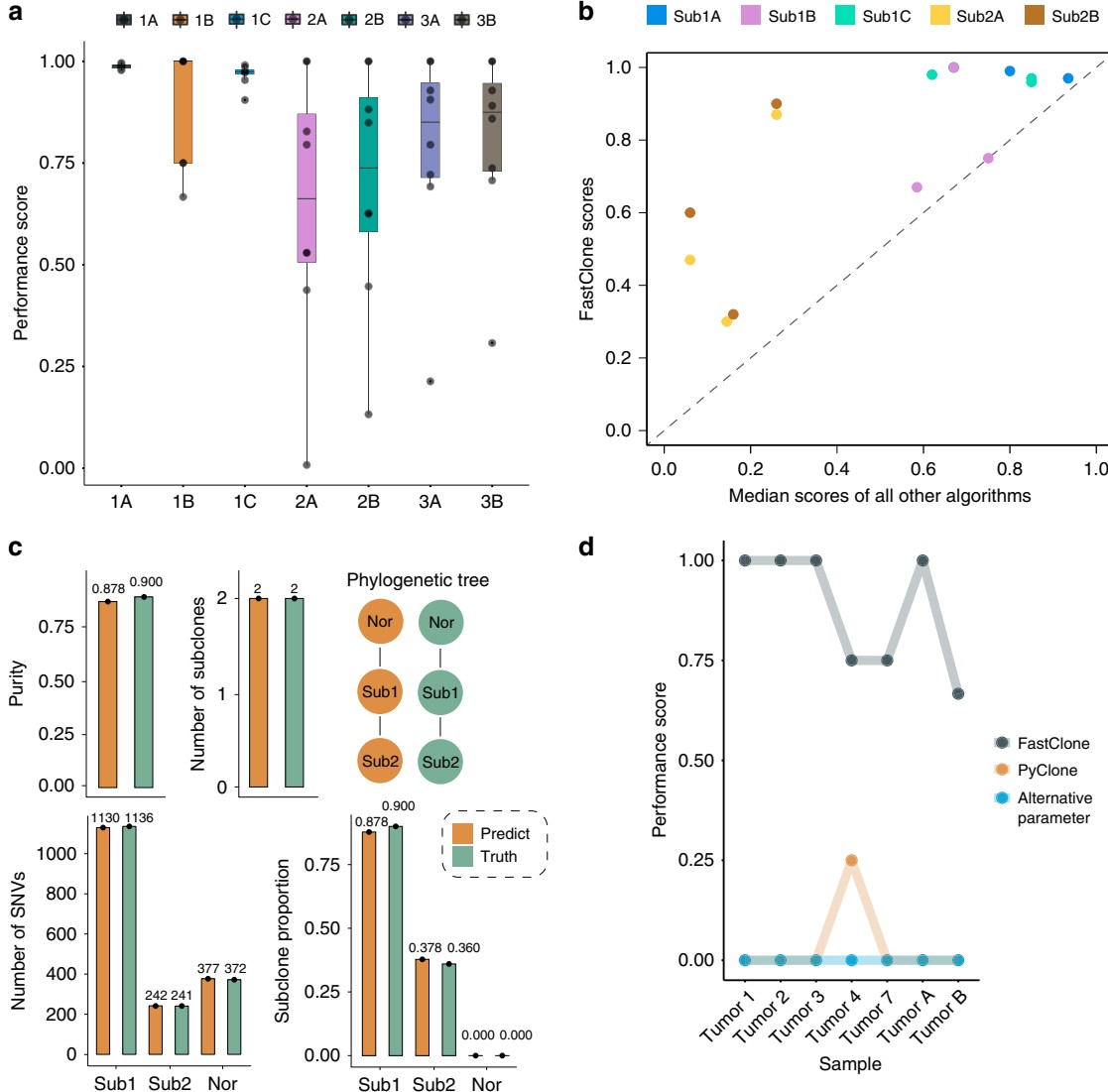

**Fig. 3 FastClone's performance on simulated data in DREAM challenge. a** Evaluation of prediction accuracy for 1A. Tumor purity, 1B. Number of subclones, 1C. Proportion of each subclone, 2A. Binary assignment of SNVs, 2B. Confusion matrix of co-occurrence of two SNVs in a subclone, 3A. Phylogeny tree, 3B. Tree relationship of individual SNVs. In the boxplots, black dots refer to each tumor sample, center lines refer to median performance, bounds of box refer to the first quartile and the third quartile of the data, and whiskers refer to min and max of the data. **b** Comparison of FastClone performance with other algorithms. Median scores of other algorithms are calculated from all entries of other teams on the leaderboard of each sub-challenge. Sub3A and Sub3B are not included in this comparison because there are no other entries on the leaderboard. **c** Example of inferred tumor heterogeneity versus ground truth. "Sub" refers to subclones and "Nor" refers to normal cell infusion. **d** The performance scores of FastClone and PyClone on predicting subclone numbers.

number of clones increases, and improves as more SNPs are included. However, the percentage of error is in general small. The error rate is only 0.13 when there are four subclones with only 500 *total* SNPs, 0.03 when 10,000 SNPs are available, and almost never wrong when there are only one or two subclones (Supplementary Fig. 3).

**Comparison of FastClone with PyClone on colon cancer data.** We next investigated the behavior and performance of FastClone compared to the current field standard. We applied both FastClone and PyClone[11] to data generated from a cohort of seven independent colon cancer tumors, each with one to three spatially distinct samples, for a total of 15 primary tumor samples[31] (Table 1). We will present the results of 14 samples (denoted as CP and their corresponding biological replicates denoted with T), as deep sequencing failed for the CP11 T1 sample. All the tumor

**Table 1 Summary of tumor samples.**

| ID | Stage | Tumor location | # Tumor samples |
|---|---|---|---|
| CP08 | T3N1M0 | Transverse colon | 2 |
| CP11 | T3N2M0 | Cecum | 2 |
| CP14 | T2N2M0 | Cecum | 3 |
| CP15 | T3N1M0 | Cecum | 2 |
| CP17 | T3N2M0 | Sigmoid | 2 |
| CP18 | T4N2M1 | Sigmoid | 2 |
| CP19 | T3N1M0 | Sigmoid | 2 |

samples we collected are at stage III cancer, which indicates that the cancer might have started to spread into the surrounding tissues and cancer cells already exist in the lymph nodes of that region. We found that 50% of the 14 primary tumor samples have

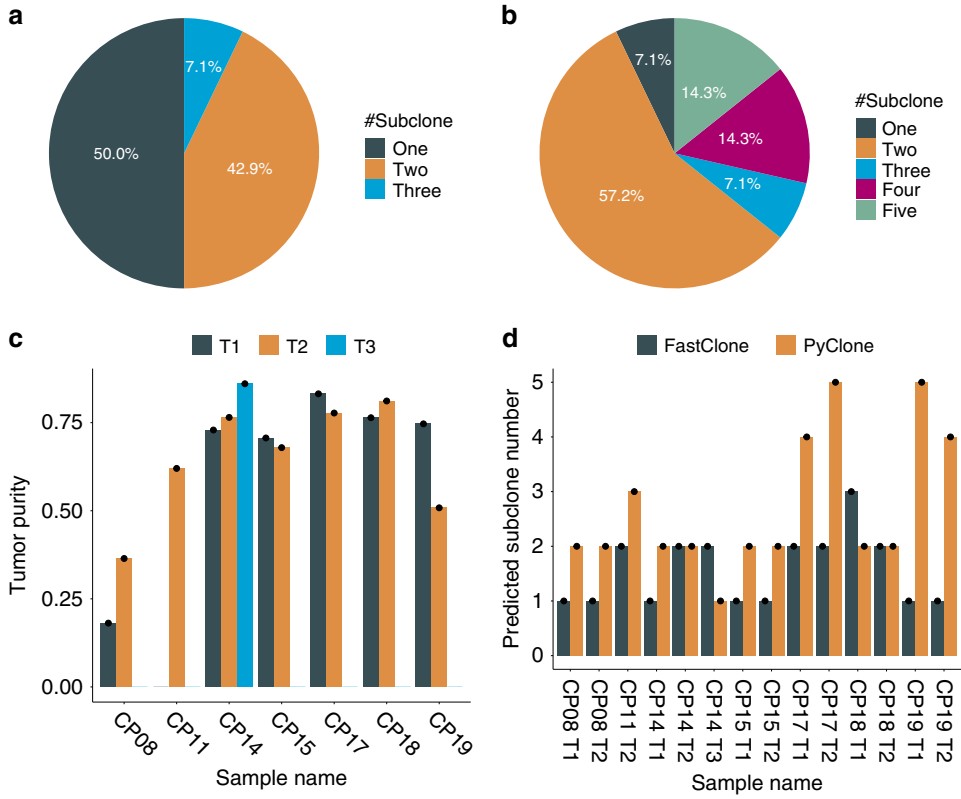

**Fig. 4 Dissection of subclones in colon cancer samples. a** FastClone's result of the distribution of single and multiple subclones among primary tumor samples. **b** PyClone's result of the distribution of single and multiple subclones among primary tumor samples. **c** The figure shows the tumor purity that is estimated by FastClone of each primary tumor sample. **d** The bar plot shows the comparison of FastClone to PyClone on predicting the number of subclones for each primary tumor sample.

multiple subclones (≥2 subclones), and only 7.1% of the 14 samples have more than two subclones (Fig. 4a).

We experimented with PyClone, a pioneering and field-standard software, on this dataset for comparison. PyClone predicted one more subclone for CP08 T1, CP08 T2, CP11 T2, and CP14 T1, as well as CP15 T1, and it predicted one less subclone for CP14 T3 as well as CP18 T1. For CP17 T1 and T2 and CP19 T1 and T2, PyClone returned many more subclones than FastClone (Fig. 4b and Supplementary Figs. 4–12). PyClone returned the same number of subclones as Fastclone for only CP14 T2, CP15 T2, and CP18 T2.

In addition, we have also compared the behaviors of FastClone and Pyclone's accuracy for each of the evaluation aspects using the eight simulated tumor data that we know the ground truth (Fig. 3d). One of the tumors (Tumor 6) did not finish estimation within a reasonable time (over a week) and thus was discarded from the analysis. Based on these comparisons, we clarified several differences between PyClone and FastClone: (1) PyClone does not estimate tumor purity; instead, it uses the tumor purity from other programs' output. (2) PyClone tends to estimate more subclones than FastClone. For example, there was one sample (simulated Tumor 1) that was predicted to have 233 subclones by PyClone using the default parameter, while the ground truth was 4. We also tuned the precision parameter of Gamma proposal function for Metropolis Hastings step to see if we could get fewer subclones, and that tumor eventually ended up with 40 subclones, but this effect was not consistent across all the samples. Some samples resulted in more subclones. Thus, we conclude that PyClone tends to predict more subclones than FastClone in general.

We further evaluated the computing speed of the algorithms in both experimentally collected tumors and simulated data. In the real-world data collected at Michigan Medicine, FastClone required no >3 s to process a single primary tumor sample (Fig. 5a, b), while PyClone takes >4 min to handle a cancer sample, which contains two or three primary tumor samples (it is preferable that PyClone uses a set of inputs to produce a valid result). Simulated data allow the number of variants to go up to tens of thousands, and we observed that as the number of variants increases, FastClone's processing time increases linearly, which is a good property for large-scale data processing (Fig. 5b). The correlation between SNV number and processing time of FastClone and PyClone is shown in Supplementary Fig. 13. This significant improvement in computation speed promises large-scale application of this software in data collections such as CancerTracer[36]. Besides PyClone, we also compared the speed and behavior of FastClone with CITUP[37]. CITUP took the longest time to analyze samples, a potential reason is that CITUP's focus is not on speed optimization. The results of CITUP show that the most likely subclonal composition of the colon cancer samples are bi-subclones, as all samples' phylogeny trees with the highest likelihood contain only two nodes. These results are more similar to the results given by FastClone than that of PyClone, which tends to report more clones.

In addition, for demonstration purposes, we also tried to extract the SNV-associated genes that only exist in multiple-subclone samples for gene set enrichment analysis (Supplementary Tables 2–4 and Supplementary Fig. 14), and found shared genes across seven multiple-subclone samples. It turns out that two variant-associated genes *KRAS* and *APC* exist in five out of seven primary tumor samples and six out of seven primary tumor samples, respectively. Moreover, three variant-associated genes *CDK19*, *TP53*, and *NLRP10* each exist in four out of the seven

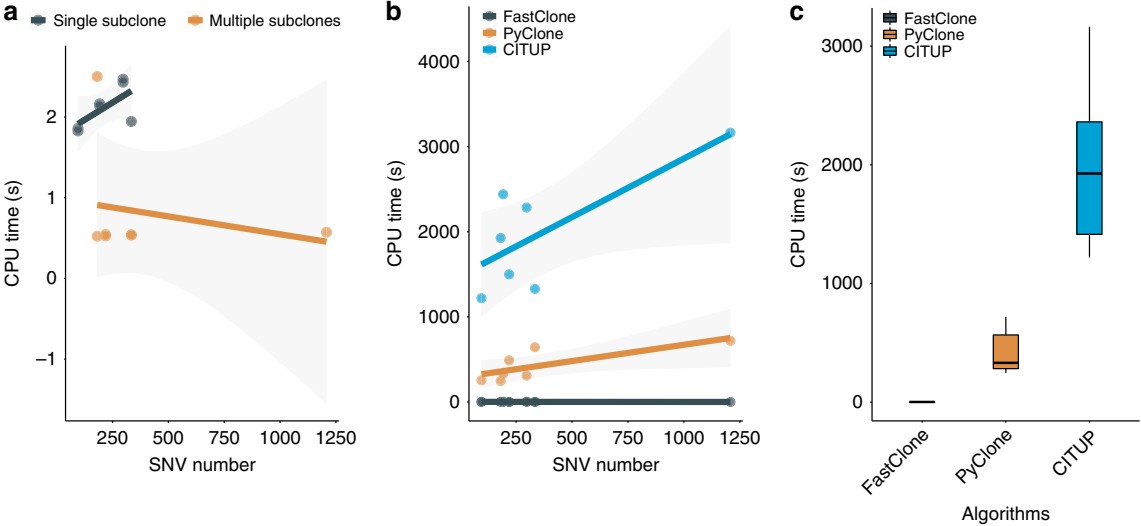

**Fig. 5 Computation time of FastClone, PyClone, and CITUP. a** Running FastClone on the patient data. The SNV number barely affects computation time in real-world data since FastClone removes noise from data first. **b** Running FastClone, PyClone, and CITUP on the patient data. Each dot at the right-bottom section represents a primary tumor sample. In general, computation time increases as the sample involves more mutations. **c** Comparing FastClone to PyClone and CITUP based on their computation time. In the box plot, center lines refer to median CPU time, bounds of box refer to the first quartile and the third quartile of the data, and whiskers refer to min and max of the data.

primary tumor samples, with no intersection among all seven primary tumor samples (Supplementary Fig. 15).

## Discussion

In this work, we present FastClone, an algorithm for modeling heterogeneity and phylogenetic trees in tumor samples. Compared with current state-of-the-art methods, FastClone features high accuracy and speed in estimating the proportion of cells with SNV and the number of subclones and assigning the SNVs to each subclone, using copy number profiles and allele frequencies as inputs. The phylogenetic tree is reconstructed subsequently by exhaustively exploring all branching possibilities. In addition, FastClone can analyze a single sample with tens of thousands of mutations within seconds, indicating that FastClone can potentially be used to assist tumor heterogeneity inference in large dataset collections.

We have demonstrated the vast reduction of runtime requirement of such analysis. We attribute this improvement to the initial approximation of the allelic frequency distribution. The traditional sampling approach requires huge numbers of iterations and often struggles to converge to agreeable clustering results. The initial approximation of the allelic frequency distribution gives us a good guess to start with, which is refined later by more precise statistical modeling.

We compared FastClone with PyClone, one gold standard algorithm in tumor heterogeneity inference in colon cancer samples. In all but two instances, FastClone inferred fewer subclones than PyClone. This does not conclude which one is better, but the two algorithms do provide alternatives of different granularity of subclone inference. FastClone evaluates tumor samples individually, whereas methods like PyClone function best when multiple, distinct but related samples are provided, such as spatially distinct samples from the same tumor or temporally assayed samples. Tools like PyClone rely on this inter-sample information when clustering groups of variants to infer subclones, performing sub-optimally if only single samples are available. We therefore view FastClone as complementary to tools such as PyClone, by focusing on subclonal inference of single samples and automatically inferring tumor purity. Furthermore, like many other tumor heterogeneity tools, since the estimation of FastClone depends on chromosomal copy

number estimations provided by upstream tools, its performance would be affected if the inferred CNA profiles by these tools are drastically different or mostly wrong across many chromosomes. However, this is unlikely to happen and thus unlikely to affect subclone identification as we are using clustering and majority vote.

Due to FastClone's substantial improvement on the processing speed, it will allow application to very large dataset to further explore the genetic markers related to multiple subclones. In the new era of precision and personalized medicine, a key question is how to stratify patients based on their genomic profiles[38–40]. We anticipate that the application of FastClone will not only benefit large-scale mechanistic study of the development of tumor heterogeneity in cancer research, but also facilitate treatment stratification in clinical settings.

## Methods

**Performance evaluation of the algorithm with in silico data.** The benchmark evaluation is performed using data from the recent ICGC-TCGA DREAM Somatic Mutation Calling-Tumor Heterogeneity Challenge[28]. The challenge organizers specified several tree structures and their subclonal compositions to represent a diverse sampling of different scenarios of tumor heterogeneity. Briefly, they used real-world sequencing data as the basis for the read generation. The reads were first aligned using BWA[41]. The results were then analyzed using Battenberg to extract the copy number profiles. The challenge organizers simulated a panel of mutations for each simulated dataset and inserted them into the sequencing reads using BAMSurgeon[32]. The modified reads were then processed using MuTect[42] to call somatic mutations. This process provides the MuTect reports and copy number profiles to predict the subclonal structures.

The predictions are then evaluated for the following aspects:

(1) Tumor purity, the proportion of tumor cells among all cells in the collected sample. The accuracy of the prediction of tumor purity is evaluated by one minus the absolute difference between true and inferred purity, a score of 1 indicates perfect performance.

(2) The number of subclones is evaluated based on relative error $s$ as

$$s = 1 - \frac{|n_{\text{truth}} - n_{\text{prediction}}|}{n_{\text{truth}}}, \tag{25}$$

where $n_{\text{truth}}$ and $n_{\text{prediction}}$ are the simulated and predicted number of subclones, respectively.

(3) Prediction of subclone proportions, which is measured by one minus the mean absolute difference between the true and inferred distribution of cellular prevalence, a score of 1 indicates perfect performance.

(4) The mutation assignment, which is evaluated based on the correlation of the co-clustering matrices. The false-positive mutations called by MuTect are

excluded from the assessment. The co-clustering matrix $S$ of the remaining mutations are calculated as

$$S = PP^{\mathrm{T}}, \tag{26}$$

where $P$ is the probability matrix of each mutation associated with each subclone ($n \times C$). The correlation between $S_{\mathrm{prediction}}$ and $S_{\mathrm{truth}}$ is the score. The subclonal composition is evaluated based on the correlation of the predicted and simulated prevalence of cells carrying each mutation.

The phylogeny, which is evaluated based on the correlation of the ancestor matrices. The false-positive mutations called by MuTect are excluded from the assessment. The ancestry matrix $M$ of the remaining mutations are calculated as

$$M = PAP^{\mathrm{T}}, \tag{27}$$

where $A$ is the asymmetric ancestor matrix of subclones ($C \times C$).

**Colon cancer tumor collection protocol**. The data utilized were obtained from Hardiman and co-workers[43]. Briefly, a total of 15 spatially distinct samples were collected from seven primary stage III colon cancer tumors. Each cancer sample provided two or three primary tumor samples, that is, at the origin of the cancers. Illumina sequencing was performed in two rounds: first via WES (Roche/NimbleGen SeqCap EZ v3), and second via custom gene panels (Agilent SureSelect XT) targeting the somatic variants detected in the initial sequencing round for each sample, the latter achieving an average read depth of 500×. The variant calls utilized for analysis were generated by at least two-caller consensus between three somatic variant callers: MuTect v.1.1.4, VarScan somatic v.2.3.7, and Strelka v.1.0.14. Copy number profiles were generated for each of the samples using Affymetrix (Thermo Fisher) OncoScan v3 SNP Arrays, resulting in profiles of 50–300-kb resolution across the genome for each tumor sample. Copy number data were processed using the Nexus Copy Number software v7.5 (BioDiscovery, El Segundo, California) using their SNP-FASST2 algorithm for analysis and segmentation, generating a median Log 2 ratio and a median B-allele frequency for each genomic segment. The segmented copy number data were additionally translated into copy number genotypes (with major and minor allele counts) in a semi-automated manner using the TAPS tool in the Patchwork software library[44].

All the experiments that were performed with FastClone can be repeated by downloading the colon data, and following the step-by-step "Supplementary Note 1. Instruction of FastClone".

**Reporting summary**. Further information on research design is available in the Nature Research Reporting Summary linked to this article.

## Data availability
The simulated tumor benchmark data are provided by the SMC Tumor Heterogeneity Challenge. Data are available at https://guanfiles.dcmb.med.umich.edu/FastClone/. The colon data we analyzed in the paper are available at https://github.com/GuanLab/FastClone_GuanLab/tree/master/colon_data. The original colon tumor variant data are available at the European Variation Archive [https://www.ebi.ac.uk/eva/?eva-study=PRJEB23791], and the SNP array data are available at GEO [https://www.ncbi.nlm.nih.gov/geo/query/acc.cgi?acc=GSE107225]. The remaining data are available in Supplementary Information or from the authors upon request.

## Code availability
The source is available at https://github.com/GuanLab/FastClone_Guanlab. To enable the wide application of this tool, FastClone_Guanlab is integrated into pip standard package installation in Python 3. FastClone code is included in Supplementary Software 1.

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

## Acknowledgements

This work is supported by NIH R35GM133346 and NSF#1452656. We thank Karin Hardiman, M.D., Ph.D. (University of Alabama, Birmingham) for use of the colorectal tumor data.

## Author contributions

H.Z. and Y.G. developed the top-performing entry to DREAM Somatic Mutation Calling-Tumor Heterogeneity Challenge. Y.X., Y.G., and H.Z. analyzed simulated data and packed the software. Y.X. and P.U. analyzed patient data. X.W., Y.X., Y.G., H.Z., and H.L. wrote the manuscript. All authors read and approved the manuscript.

## Competing interests

The authors declare no competing interests.
