## [Peer Review File · Nature Communications]

REVIEWER COMMENTS

Reviewer #1 (Remarks to the Author): Expert in computational genomics

Xiao et al report a new computational tool, FastClone, for deciphering tumor heterogeneity from bulk sequencing data. There are already a number of tools that perform this task. However, this paper reports that FastClone demonstrated very good performance in the SMC-Het DREAM Challenge compared to other methods. While the DREAM Challenge used data simulated by BAMSurgeon, this paper also compares the results from FastClone on a 'real' metastatic colon cancer dataset with those from a commonly used previously developed tool, PyClone.

Overall, the FastClone method is described well. However, further detail is required in places:

- p is used throughout the paper, but is not defined.
- As stated in the manuscript, the cellular prevalence of a mutation in a region that has had a copy number gain depends on whether the mutation has occurred before or after the gain, and it is necessary to use different equations for each scenario. However, in general the order of occurrence of genomic events is not known. How does FastClone decide which equation to use?
- Related to the point above, it is not unusual for multiple gains to occur in the same region of the genome leading to copy numbers states such as 3+1, 4+1, etc. In this situation, mutations may occur before some gains but after others. Does FastClone allow for this possibility? If not, it's not fair to describe the paper as 'a comprehensive analysis of different scenarios'.
- Is there some biological reason for believing that clones containing more SNVs will be towards the leaves of the phylogenetic tree or is this a purely arbitrary choice?
- The colon cancer data reported come from a paper that used CITUP to decipher heterogeneity. It would therefore seem natural to compare FastClone with this method. The authors should justify their decision instead to compare FastClone with PyClone.

Minor comments:

- Stating that targeted therapy is 'likely' to elicit an 'impressive' initial response seems like an overstatement.
- There are several reasons why it's difficult to obtain longitudinal samples from solid cancers in addition to economic ones, notably (a) the ethics of carrying out unnecessary invasive procedures (b) practicalities: the majority of cancer samples are obtained from surgical procedures. If a cancer has been removed at one timepoint it will not, by definition, be available for sampling at a later timepoint.

Reviewer #2 (Remarks to the Author): Expert in computational genomics

Intra-tissue heterogeneity is a bottleneck in comprehending cancer mechanisms and therapeutic strategies. Several methods were developed to deconvolute tumor heterogeneity, yet the issue has not been fully addressed. Xiao et al developed FastClone to deconvolute heterogeneity in bulk-sequenced samples. Authors have demonstrated the efficiency of FastClone on colon cancer and simulation data. Overall, the methodology is technically sound. However, upon reviewing carefully I have following comments. Addressing these points will facilitate a better understanding and further establish the merits.

Major concerns:

1. The estimation of FastClone depends on the CNA values. How to get the chromosomal copy number estimation? Chromosomal copy number estimations from different tools may be different. Will FastClone be affected? Can FastClone estimate copy number?
2. Authors assumed two different situations about 1-state CNA events, SNV happens after CNA or before. For CNA happens before an SNV, Fig. 2a4-1 shows a simple case that allelic copy number

equals 2. However, it is common that CNA event happens multiple times before an SNV, or would be amplified again after SNV. That means SNV not necessarily happens to one allele. Thus, the prevalence of cell with SNV would be over- or under-estimated through eq.5 or eq.9. The estimation of p is bigger than 1 may be caused by this reason.

3. Authors generated 8 simulation data to evaluate the performance of FastClone from three aspects. How about the power of other tools on the simulation data? The comparison would help to assess if FastClone has the advantage or not. What's the relationship between the number of subclones and estimation power?

4. Would the number of SNVs affect FastClone estimation? How many SNVs at least are required?

5. Although authors compare FastClone and Pyclone on real dataset, the ground truth are unknown. To fully convince readers if FastClone offers improved power in tumor deconvolution, it would be more appropriate to compare with traditional analysis methods by analyzing datasets where the true heterogeneity information is known.

6. It would be good to provide an overview of FastClone as a figure in the maintext.

7. The FastClone can not be successfully installed on my Mac. Authors should check if the FastClone can be successfully installed to use. Additionally, what format of input data should be prepared? More details about FastClone usage should be described. It would be also useful to provide example output files so that the users know what to expect

Minor concerns:

1. In Eq.5,7 and 9, what is q ? It is also should be p ?

2. Authors used different symbols to describe the same terminology. For example, p, p, w are used to represent the proportion of subclone. That will cause confuse.

3. The Eq. 21 and 24 do not followed standard format of Binomial distribution. It should be $\text{Binom}(m_k, r_k; \beta)$ instead of $\text{Binom}(\beta_k, r_k; \beta)$.

REVIEWER COMMENTS

Reviewer #1 (Remarks to the Author): Expert in computational genomics

Xiao et al report a new computational tool, FastClone, for deciphering tumor heterogeneity from bulk sequencing data. There are already a number of tools that perform this task. However, this paper reports that FastClone demonstrated very good performance in the SMC-Het DREAM Challenge compared to other methods. While the DREAM Challenge used data simulated by BAMSurgeon, this paper also compares the results from FastClone on a ‘real’ metastatic colon cancer dataset with those from a commonly used previously developed tool, PyClone.

Overall, the FastClone method is described well. However, further detail is required in places:

Thank you.

- ρ is used throughout the paper, but is not defined.

Thank you for pointing this out. Its definition is included now on page 2 (the symbol was changed to ϱ , respecting the feedback in the second review) “the prevalence of cells that contain a certain SNV in the tumor sample”.

To make all symbols clearer to the readers, we have now added a symbol table (Supplementary Table S1) and copied below..

Table S1: Symbols and their definitions

Symbol	Definition
ϱ	the prevalence of cells that contain a certain SNV in the tumor sample
ρ_j	the prevalence of cells that contain the j -th SNV in the tumor sample
β	allele frequency of a certain SNV
β_j	allele frequency of the j -th SNV
$\hat{\beta}_{jk}$	expected allele frequency of the k -th mutation if it is associated with the j -th subclone
N_{major}	major copy numbers of CNA

N_{minor}	minor copy numbers of CNA
n_{cell}	total cell number in the tumor
n	total number of SNVs
Z	normalization constant
h	bandwidth for smoothing density estimations
d	number of samples
C	total number of subclones
w_j	the proportion of the j -th subclone
L_{jk}	the probability of the k -th mutation associated with the j -th subclone
λ	log-likelihood of the entire mutation assignment
m_k	observed reads that carry the k -th mutation
r_k	total number of reads that cover the locus of the k -th mutation and pass the quality filter

• As stated in the manuscript, the cellular prevalence of a mutation in a region that has had a copy number gain depends on whether the mutation has occurred before or after the gain, and it is necessary to use different equations for each scenario. However, in general the order of occurrence of genomic events is not known. How does FastClone decide which equation to use?

Thank you for this excellent question. FastClone cannot decisively decide which equation to use, maybe nor can any other algorithms, if applied to bulk tumors. However this ambiguity does not completely impede the estimation of subclones and assignment of SNVs. FastClone calculates association scores between SNVs and subclones for every scenario separately and considers them of equal probability, and assigns the SNVs to the one with the highest association score among all scenarios. This association score will be different for each scenario, because if the scenario is correct, the q will be more likely assigned to that peak. During estimation of the number of clones, all scenarios get votes if there are no sufficient SNVs on the chromosomes without CNA (i.e. <100, the default). Though individually some of the scenarios are wrong, statistically across many SNVs, this can help us to generate the correct number of clones, especially together with the SNVs on chromosomes without CNAs.

This point has been clarified on Page 3, and has also been illustrated by Figure 1-d.

“Since one cannot directly determine the order of occurrence of genomic events and decide which set of equations to use, FastClone calculates the q value of every possible scenario. Then, the association scores between SNVs and subclones for all scenarios are calculated separately, and the SNVs are assigned to the subclone with the highest association score among all scenarios. This association score will be different for each scenario, because if the scenario is correct, the q will be more likely to be assigned to that peak.

During the estimation of the number of clones, all scenarios get votes if there are no sufficient SNVs on the chromosomes without CNA (*i.e.*, <100 SNVs). Though individually some of the scenarios are wrong, statistically across many SNVs, this can help us to generate the correct number of clones, especially together with the SNVs on chromosomes without CNAs. The steps of estimating subclone numbers and assigning SNVs to subclones would be described in detail in the next two sections.”

Figure 1. Overview of the FastClone algorithm. **(a)** The tumor sample is heterogeneous, composed of both normal cells as well as tumor subclones (top panel). The tumor dynamically evolves throughout the disease course, generating subclones with different genotypes (bottom panel). The dots in different colors represent different SNVs. **(b)** The DNA sequencing of the bulk tumor provides information about 1) allele frequency of each SNV (β), *i.e.*, the observed allele occurrence among all cells (top panel); 2) a CNA profile in the form of N_{major} and N_{minor} (bottom panel). Each of the yellow dots represents a copy of the allele with a certain SNV. N_{major} represents the number of the most frequent copy of an allele. Counterwisely, N_{minor} represents the number of the less frequent copy of the allele. **(c)** FastClone Model. First, we calculated the proportion of cells that carry each SNV (ρ_i for SNV_{*i*}) based on the allele frequency and CNA profile. This calculation is discussed in two situations: with and without CNA events. Under the situation of ‘With CNA Events’, multiple possibilities are further discussed (see Figure 2). Then, subclone numbers, subclone proportions, and tumor purity are determined from the distribution of

q. After that, SNVs are assigned to subclones. Finally, the putative evolutionary relationship of the subclones is established. **(d)** The workflow of FastClone algorithm. The workflow starts with sequencing information as input, which includes allele frequency (β) and CNA profile (N_{major} and N_{minor}). Since we do not know the order of occurrence of genomic events, all possible scenarios are discussed, and q value for each scenario is calculated separately. The number of possible scenarios equals the value of N_{major} . Then KDE is used to determine the distribution of q . Each peak in the q distribution indicates a subclone. After that, association scores between each SNV-subclone pair are calculated. Then the SNV is assigned to the subclone with the highest association score. If there are several q values associated with one SNV, then the q that provides the highest association score is used to assign the SNV to subclone (in this case, q_2 that assigns this SNV to the green subclone is considered the correct solution). Finally, the most likely phylogeny tree of the subclones is constructed.

- Related to the point above, it is not unusual for multiple gains to occur in the same region of the genome' leading to copy numbers states such as 3+1, 4+1, etc. In this situation, mutations may occur before some gains but after others. Does FastClone allow for this possibility? If not, its not fair to describe the paper as 'a comprehensive analysis of different scenarios'.

Thank you very much for communicating this concern. We did not consider these possibilities. It was a defect inherited from how we conducted from the initial challenge. We have made corrections to accommodate these situations you mentioned. Please see equation update on Page 4 and 6, Equation 5-8, 11-12, 18-20 & 22-23, as well as Figure 2a,2b. Relevant experiments have been rerun and result updated. The inferred number of clones was not changed in any of the samples.

Figure 2a, 2b. Gray bars represent maternal chromosomes and brown bars represent paternal chromosomes, blue spheres represent normal loci, and red spheres represent mutated loci that contain SNVs.

- Is there some biological reason for believing that clones containing more SNVs will be towards the leaves of the phylogenetic tree or is this a purely arbitrary choice?

This is an at-best guess based on some biological reasoning. As cancer progresses, there is likely to be an accelerated increase of mutations, due to loss of DNA damage repair functions. It is often seen that DNA

damage repair genes are mutated in tumors, and thus leads to more SNVs as the tumor progresses (*i.e.*, towards the leaves). This has been explained on Page 7 :

“This assumption has its biological reason that as cancer progresses, the mutation rate is likely to be accelerated due to accumulated mutations in key genome stability pathways, such as DNA repair and replication pathways or cell-cycle checkpoints (Jackson and Loeb 1998; Peterson and Kovyrshina 2017).”

- The colon cancer data reported come from a paper that used CITUP to decipher heterogeneity. It would therefore seem natural to compare FastClone with this method. The authors should justify their decision instead to compare FastClone with PyClone.

Thank you. We compared FastClone to CITUP, and it turned out that CITUP took the longest time to analyze a sample. The results of CITUP show that the most likely subclonal composition of our real samples are bi-subclones, as all samples' phylogeny trees with the highest likelihood contain only 2 nodes. We have now added this comparison of the result between FastClone and CITUP on Page 9 and Figure 5-b,c.

“Besides PyClone, we also compared the speed and behavior of FastClone with CITUP. CITUP took the longest time to analyze samples, a potential reason is that CITUP's focus is not on speed optimization. The results of CITUP show that the most likely subclonal composition of the colon cancer samples are bi-subclones, as all samples' phylogeny trees with the highest likelihood contain only 2 nodes. These results are more similar to the results given by FastClone than that of PyClone, which tends to report more clones.”

Since CITUP always produces two subclones in the samples we analysed, we did not include a separate figure for the number of clones (other than text) for CITUP.

Figure 5b, 5c. (b) Running FastClone, Pyclone and CITUP on the patient data. Each dot at the right-bottom section represents a primary tumor sample. In general, computation time increases as the sample involves more mutations. **(c)** Comparing FastClone to PyClone and CITUP based on their computation time.

Minor comments:

- Stating that targeted therapy is ‘likely’ to elicit an ‘impressive’ initial response seems like an overstatement.

Thank you. We have now edited the statement to ‘might have the potential to elicit promising initial-responses’ (Page 1).

- There are several reasons why it’s difficult to obtain longitudinal samples from solid cancers in addition to economic ones, notably (a) the ethics of carrying out unnecessary invasive procedures (b) practicalities: the majority of cancer samples are obtain from surgical procedures. If a cancer has been removed at one timepoint it will not, by definition, be available for sampling at a later timepoint.

Thank you for this great suggestions. We have added this section (Page 1):

“However, this strategy may not be suitable under all circumstances because of the ethics of carrying out unnecessary invasive procedures as well as the practicality of longitudinal sampling for solid tumors, since the majority of cancer samples are obtained from surgical procedures, and if a tumor has been removed at the one-time point, it will not be available for sampling at a later time point.”

Reviewer #2 (Remarks to the Author): Expert in computational genomics

Intra-tissue heterogeneity is a bottleneck in comprehending cancer mechanisms and therapeutic strategies. Several methods were developed to deconvolute tumor heterogeneity, yet the issue has not been fully addressed. Xiao et al developed FastClone to deconvolute heterogeneity in bulk-sequenced samples. Authors have demonstrated the efficiency of FastClone on colon cancer and simulation data. Overall, the methodology is technically sound. However, upon reviewing carefully I have following comments. Addressing these points will facilitate a better understanding and further establish the merits.

Thank you for your time in reviewing this article. We have addressed your comments below:

Major concerns:

1. The estimation of FastClone depends on the CNA values. How to get the chromosomal copy number estimation? Chromosomal copy number estimations from different tools may be different. Will FastClone be affected? Can FastClone estimate copy number?

Chromosome copy number variation in the colon cancer samples is obtained by Nexus Copy Number software v7.5 . FastClone does not estimate copy number, instead, it ingests these copy number genotypes in addition to the variant allele frequencies; it does not generate them *de novo*. In simulated data related to the challenge, the CNV values are obtained by applying BAMSurgeon to an already-sequenced tumor cell line, which is sequenced by SAMtools.

These methods are included on page 7.

“These simulated samples were based on the already-sequenced tumor cell line, which is sequenced by SAMtools³⁴, and their chromosome copy number variations were obtained by Nexus Copy Number software v7.5.”

As you mentioned, many tools can estimate CNA. Yes, if the inferred CNA profiles are drastically different/mostly wrong across many chromosomes, certainly FastClone will be vulnerable. But if the majority of the CNA inference are correct, it is unlikely to affect subclone identification as we are using practically clustering and majority vote. This has now been discussed on Page 10.

“Furthermore, like many other tumor heterogeneity tools, since the estimation of FastClone depends on chromosomal copy number estimations provided by upstream tools, its performance would be affected if the inferred CNA profiles by these tools are drastically different or mostly wrong across many chromosomes. However this is unlikely to happen and thus unlikely to affect subclone identification as we are using clustering and majority vote.”

2. Authors assumed two different situations about 1-state CNA events, SNV happens after CNA or before. For CNA happens before an SNV, Fig. 2a4-1 shows a simple case that allelic copy number equals 2. However, it is common that CNA event happens multiple times before an SNV, or would be amplified again after SNV. That means SNV not necessarily happens to one allele. Thus, the prevalence of cell with SNV would be over- or under-estimated through eq.5 or eq.9. The estimation of ρ is bigger than 1 may be caused by this reason.

Thank you for this clear description of your concern. This is indeed an insightful point. We did not consider the possibility of a three-step or more scenario -- it was a defect. We have now made corrections to code to accommodate these situations you mentioned. Please see equation update on Page 4 and 6, Equation 5-8, 11-12, 18-20 & 22-23, as well as Figure 2a,2b. Relevant results have been rerun and updated. However in no cases the inference of the number of clones was changed.

Thank you very much for your expertise in making this a more solid piece of work.

Figure 2a, 2b. Gray bars represent maternal chromosomes and brown bars represent paternal chromosomes, blue spheres represent normal loci, and red spheres represent mutated loci that contain SNVs.

3. Authors generated 8 simulation data to evaluate the performance of FastClone from three aspects. How about the power of other tools on the simulation data? The comparison would help to assess if FastClone has the advantage or not. What's the relationship between the number of subclones and estimation power?

We have now included a comparison of FastClone against other 5 tools on the 8 simulated data. Additionally, we carried out a comparison of FastClone versus PyClone on the simulated data (presented in answering your question 5). Overall, FastClone performs better than the other tools in a diverse set of evaluations. This has been explained on Page 7 and 8.

“Then, we compared the performance of FastClone against other programs.”

“Sub-challenge 1A evaluates accuracy in predicting tumor purity.” “On the challenge leaderboard, we procured the same best result with another algorithm with the median performance score of 0.99, which is an excellent performance for simulated tumor samples (Fig. S1). The performance scores of all other 4 software range from 0.37 to 0.89, with a median score of 0.77 (Fig. 3b)³⁵.”

“Sub-challenge 1B evaluates the accuracy in predicting the number of subclones” “On the leaderboard, FastClone ranked second place among all software with the median score of 0.75 (Fig. S1), and the scores of the other 5 software range from 0.38 to 1, with a median score of 0.67 (Fig. 3b)³⁵.”

“1C evaluates the accuracy in predicting the proportion for each subclone.” “On the leaderboard, FastClone obtained a score of 0.97 (Fig. S1). The scores of the other 5 software range from 0.6 to 0.89, with a median score of 0.74 (Fig. 3b)³⁵.”

“Sub-challenge 2 evaluates the performance of the algorithm on determining mutation assignments to subclones.” “The leaderboard median scores of 0.47 and 0.6 place FastClone at the top in this sub-challenge (Fig. S1)³⁵, and the scores of the other 5 software (for both sub-challenge 2A and 2B) range from 0.09 to 0.47, with a median score of 0.21 (Fig. 3b).”

“Sub-challenge 3 focuses on evaluating the prediction accuracy of subclone phylogeny” “For sub-challenge 3A, we obtained a median score of 0.69, being the only team with a final submission on the leaderboard. There are no entries on the leaderboard for sub-challenge 3B.”

“Overall, FastClone has an excellent performance in predicting the tumor purity, the number of subclones, and the proportion of each subclone. In almost all sections, FastClone had the highest median score among all the models participated (Fig. S1, Fig. 3b)³⁴. This outstanding performance in SMC-Het Challenge suggests that FastClone is a state-of-the-art model in the field of tumor subclone reconstruction.”

Figure S1. Performance comparison among submissions. Median performance scores of different algorithms in each sub-challenge. Sub1A predicts purity of the tumor. Sub1B predicts number of subclones. Sub1C predicts subclone proportions. Sub2A and Sub2B predict mutation assignments to subclones from different perspectives. Sub2A predicts the assignments of each SNV to subclone, and Sub2B predicts the probabilistic clustering of SNVs. SMC-Tester is the PhyloWGS baseline.

Figure 3b. Comparison of FastClone performance with other algorithms. Median scores of other algorithms are calculated from all entries of other teams on the leaderboard of each sub-challenge. Sub3A and Sub3B are not included in this comparison because there are no other entries on the leaderboard.

Additionally, in the revision, we compared the performance and behavior with more depth with CITUP (Page 9 and Figure 5b. c). CITUP takes exponentially more time than FastClone. In all the samples in the colon tumor sample, the highest probability tree in CITUP reported two clones.

“Besides PyClone, we also compared the speed and performance of FastClone with CITUP. CITUP took the longest time to analyze samples, a potential reason is that CITUP’s focus is not on speed optimization (Fig. 5c). The results of CITUP show that the most likely subclonal composition of our real samples are bi-subclones, as all samples’ phylogeny trees with the highest likelihood contain only 2 nodes. These results are more similar to the results given by FastClone than that of PyClone, which tends to report more clones.”

Another difference between PyClone and FastClone is that the former does not estimate purity, but treats the sample as a pure tumor, this has been emphasized on Page 9.

“Based on these comparisons, we clarified several differences between Pyclone and FastClone: 1) Pyclone does not estimate tumor purity, instead, it uses the tumor purity from other programs’ output. 2) Pyclone tends to estimate more subclones than FastClone.”

Figure 5b, 5c. (b) Running FastClone, PyClone, and CITUP on the patient data. Each dot at the right-bottom section represents a primary tumor sample. In general, computation time increases as the sample involves more mutations. **(c)** Comparing FastClone to PyClone and CITUP based on their computation time.

Thank you very much for the question of the relationship between the number of subclones and the accuracy of estimating the number of subclones. Empirically from simulated data we observe a drop when the number of subclones increases, though it is certainly affected by complicated factors such as the number of SNPs, average reads of each SNP, as well as the distribution of q s of the clones. As such, in response to this question, we used 100 simulations in each case to assess the relationship between the number of SNPs, and the number of subclones, with randomized q distributions to cover as many possibilities as possible. As expected, the accuracy drops as the number of clones increase, and improves as more SNPs are included. The percentage of error is in general small, for only 13% rate when there are four subclones with only 500 *total* SNPs. This error happens when two clone's q s are too close to each other. This has now been added into Supplementary Figure S3 and text on Page 8:

“Furthermore, we assessed how the number of subclones and the number of SNVs affect FastClone’s accuracy of estimating the number of subclones through simulation experiments. As expected, the accuracy drops as the number of clones increases, and improves as more SNPs are included. However, the percentage of error is in general small. The error rate is only 0.13 when there are four subclones with only 500 *total* SNPs, 0.03 when 10,000 SNPs are available, and almost never wrong when there are only one or two subclones (Supplementary Figure S3).”

Figure S3: FastClone performance on different numbers of subclones. We simulated q distributions with different numbers of subclones and different numbers of SNVs from hypergeometric distribution. The q values were sampled from a uniform distribution between 0 and 1. For each case, we performed 100 simulations and used FastClone to predict q values. Then the average error rate per subclone was calculated. As the number of SNVs increased, the error rate dropped and approached zero.

4. Would the number of SNVs affect FastClone estimation? How many SNVs at least are required?

Thank you for your question. We randomly subsampled the SNVs of the 8 simulated data (for which we do know the gold standard), and observed the changes of the predictions for purity and number of clones. For each tumor, we randomly sampled 99 times: 0.01, 0.02, ... to 0.98, 0.99 of the original number of SNVs. Although SNV is not the only influential factor (as discussed above) affecting performance, overall we found errors are rare for most tumors, even with only dozens of SNVs. For the two tumors (Tumour 7 and Tumour B), which we do find a change in predicted number of clones, the clones that were dropped off are the ones containing the least SNVs, for which the sampling was not sufficient. For example, Tumour 7, with a total of 2834 SNVs, had a clone of only 89 SNVs (3% of total population), when we subsample it to 200 SNVs, we have about 9 SNVs left, and this number was not statistically strong enough to support a separate clone. We have now added this simulation conclusion to Page 8 and supplementary Figure S2.

“To investigate whether the number of SNVs would affect FastClone estimation, we randomly subsampled the SNVs of the 8 simulated tumor samples, and observed the changes of the predictions for purity and number of clones. For each tumor, we randomly sampled SNVs for 99 times: from 1 to 99 percent of the original number of SNVs. Overall we found errors are rare for most tumors even with only dozens of SNVs (Fig. S2). For the two tumors (Tumour 7 and Tumour B), which we did find a change in

the predicted number of clones, the clones that were dropped off are the ones containing the least SNVs, for which the sampling was not sufficient. For example, Tumour 7, with a total of 2834 SNVs, had a clone of only 89 SNVs (3% of the total population). When we subsampled it to 200 SNVs, we had only about 9 SNVs left, and this number was not statistically strong enough to support a separate clone. In conclusion, FastClone’s predictions are mostly robust across a wide range of SNVs in tumor samples.”

Figure S2. Number of subclones predicted by FastClone in each tumor sample is mostly robust across different numbers of SNVs. The x axis shows the number of SNVs randomly sampled from the tumor data, and the y axis shows the predicted number of subclones in the tumors.

5. Although authors compare FastClone and Pyclone on real dataset, the ground truth are unknown. To fully convince readers if FastClone offers improved power in tumor deconvolution, it would be more

appropriate to compare with traditional analysis methods by analyzing datasets where the true heterogeneity information is known.

Thank you very much for pointing this out. We have now compared the performance of FastClone and Pyclone's accuracy for each of the evaluation aspects using data that we know the ground truth (Supplementary Figure S13 and Figure 3d).

We wanted to communicate with you that one of the tumors was left out in comparison, as on our machines PyClone did not finish running for over a week (while FastClone finished within minutes). We decided to leave it out as the time is expected to be O^2 and unlikely to finish within reasonable time.

We have also clarified several differences between PyClone and FastClone: 1. PyClone does not estimate tumor purity, i.e. it assumes the entire sample is tumor without normal cell infusion. 2. PyClone tends to estimate much more clones than FastClone. For example, there was one sample that had 233 clones using the default parameter. We also tried to tune the parameters to see if we could get fewer clones, and that tumor eventually ended up with 40 clones. Again, by no means do we mean that this proves one is better than the other, as if dividing infinitely, in extreme cases, each cell might be considered as one clone. Rather, we presented this difference as part of behaviors of the softwares in the manuscript.

We have now included these explanations to Page 9, Figure 3d and Figure S13.

“Additionally, we have also compared the behaviors of FastClone and Pyclone's accuracy for each of the evaluation aspects using the 8 simulated tumor data that we know the ground truth (Fig. 3d). One of the tumors (Tumour 6) did not finish estimation within a reasonable time (over a week) and thus was discarded from the analysis. Based on these comparisons, we clarified several differences between Pyclone and FastClone: 1) Pyclone does not estimate tumor purity, instead, it uses the tumor purity from other programs' output. 2) Pyclone tends to estimate more subclones than FastClone. For example, there was one sample (simulated Tumor 1) that was predicted to have 233 subclones by Pyclone using the default parameter, while the ground truth was 4. We also tuned the precision parameter of Gamma proposal function for Metropolis Hastings step to see if we could get fewer subclones, and that tumor eventually ended up with 40 subclones, but this effect was not consistent across all the samples. Some samples resulted in more subclones. Thus, we conclude that PyClone tends to predict more subclones than FastClone in general.”

Figure S13. The correlation between SNV number and processing time of FastClone, PyClone, and PyClone with an alternative Gamma precision parameter on 8 simulated data from DREAM challenge.

Figure 3d. The performance scores of FastClone and PyClone on predicting subclone number, which are calculated by SMC-DREAM challenge's python script of scoring.

6. It would be good to provide an overview of FastCLone as a figure in the maintext.

Thank you, Figure 1-d has been edited to include the workflow of FastClone, illustrating how we estimated tumor purity, subclone number, subclone prevalence, assignment of SNVs to subclones and tumor phylogeny trees using the input of allele frequency and CNA profile.

Figure 1. Overview of the FastClone algorithm. **(a)** The tumor sample is heterogeneous, composed of both normal cells as well as tumor subclones (top panel). The tumor dynamically evolves throughout the disease course, generating subclones with different genotypes (bottom panel). The dots in different colors represent different SNVs. **(b)** The DNA sequencing of the bulk tumor provides information about 1) allele frequency of each SNV (β), *i.e.*, the observed allele occurrence among all cells (top panel); 2) a CNA profile in the form of N_{major} and N_{minor} (bottom panel). Each of the yellow dots represents a copy of the allele with a certain SNV. N_{major} represents the number of the most frequent copy of an allele. Counterwisely, N_{minor} represents the number of the less frequent copy of the allele. **(c)** FastClone Model. First, we calculated the proportion of cells that carry each SNV (ρ_i for SNV_{*i*}) based on the allele frequency and CNA profile. This calculation is discussed in two situations: with and without CNA events. Under the situation of ‘With CNA Events’, multiple possibilities are further discussed (see Figure 2). Then, subclone numbers, subclone proportions, and tumor purity are determined from the distribution of ρ . After that, SNVs are assigned to subclones. Finally, the putative evolutionary relationship of the subclones is established. **(d)** The workflow of FastClone algorithm. The workflow starts with sequencing information as input, which includes allele frequency (β) and CNA profile (N_{major} and N_{minor}). Since we do

not know the order of occurrence of genomic events, all possible scenarios are discussed, and q value for each scenario is calculated separately. The number of possible scenarios equals the value of N_{major} . Then KDE is used to determine the distribution of q . Each peak in the q distribution indicates a subclone. After that, association scores between each SNV-subclone pair are calculated. Then the SNV is assigned to the subclone with the highest association score. If there are several q values associated with one SNV, then the q that provides the highest association score is used to assign the SNV to the subclone (in this case, q_2 that assigns this SNV to the green subclone is considered the correct solution). Finally, the most likely phylogeny tree of the subclones is constructed.

7. The FastClone can not be successfully installed on my Mac. Authors should check if the FastClone can be successfully installed to use. Additionally, what format of input data should be prepared? More details about FastClone usage should be described. It would be also useful to provide example output files so that the users know what to expect

Thank you very much for checking in this. We found out the pip issue was caused by another package with the same name as ours on PyPI server during the time of the review. We have now changed the repository name to “FastClone_GuanLab” to fix the problem. We have updated some code and updated the Github and made sure it is installable in python 3.5 or later version. We have tested installation on the following platforms: macOS Catalina 10.15.3 and Ubuntu 18.04.2 LTS .The Github issue page will allow further tracking of issues if they occur. Usage of FastClone is now included in Supplementary files, and updated on GitHub:

FastClone:

1. Installation

FastClone needs Python 3.5 or later version. It needs logbook, python-fire, scikit-learn, and pandas. To install the package, please use the following codes:

```
git clone https://github.com/GuanLab/FastClone\_GuanLab.git
pip install FastClone_GuanLab/
```

(Please make sure you have the slash at the end, which forces pip to install from local directory, otherwise it will run into error)

An alternative way is using pip to directly install FastClone:

```
pip install fastclone-guanlab
```

2. Usage

FastClone accepts either MuTect VCF + Battenberg format (specified in the DREAM SMC-Het Challenge) or PyClone format. The general format of the command line for running FastClone:

```
fastclone load-[FILE_FORMAT] prop [FILE_NAME] [TUMOR_PURITY] solve
[OUTPUT_PATHWAY]
```

Here is an example to load a sample and infer:

```
fastclone load-pyclone prop t1.tsv 0.8 solve ./fastclone_result
```

(Please make sure t1.tsv is under your current directory)

Minor concerns:

1. In Eq.5,7 and 9, what is q ? It is also should be ρ ?

Thank you very much for pointing it out. We have changed all the q , ρ , and q in the manuscript to q since only this font of q can be typed in equations. Additionally, a reverse check list of all symbols is added in Table S1.

2. Authors used different symbols to describe the same terminology. For example, ρ, p, w are used to represent the proportion of subclone. That will cause confuse.

We apologize for the confusion. q and p have basically the same meaning, which is “the proportion of cells that carry a certain SNV”. But w is different from them. Here the symbol ‘ w ’ is defined as “the proportion of a certain subclone”. The difference between “cells that carry a certain SNV” and “a certain subclone” is that a subclone may contain several SNVs. In order to avoid confusion, we have changed all the ‘ p ’ to ‘ q ’, and added some descriptions of the differences between ‘ w ’ and ‘ q ’. The added descriptions are as below (Page 5):

“Note that w_j is different from ρ_j , which is defined as the proportion of cells that contain the j -th SNV, because a subclone may contain several SNVs.”

Thanks again for your advice.

3. The Eq. 21 and 24 do not followed standard format of Binomial distribution. It should be $\text{Binom}(m_k, r_k; \beta)$ instead of $\text{Binom}(\beta_k, r_k; \beta)$.

Thank you for pointing that out. The two occurrences of this problem in Eq.21 and Eq.24 have now been changed to $\text{Binom}(m_k, r_k; \hat{\beta})$ (Page 6).

REVIEWERS' COMMENTS:

Reviewer #1 (Remarks to the Author):

Overall I'm happy with the changes made. However, the wording of 2 sections is still unclear:

1. On p.6 'as we do know the exact number of alleles' should, I think, read 'as we do **not** know the exact number of alleles'.

2. The meaning of the phrase 'if there are no sufficient SNVs on the chromosomes without CNA (i.e. , <100 SNVs, the default)' is unclear. My understanding of the paragraph containing this phrase is that it refers to the order of acquisition of individual SNVs. Assigning mutations collectively to a particular timepoint (before or after a copy number gain) would be a mistake, since mutations are likely to occur throughout the evolution of a tumor, both before and after copy number changes.

Reviewer #2 (Remarks to the Author):

The authors have done a great job addressing all my previous concerns.

All responses and edits in the manuscript are marked in Blue.

All original reviews are in Black.

REVIEWER COMMENTS

Reviewer #1 (Remarks to the Author):

Overall I'm happy with the changes made. However, the wording of 2 sections is still unclear:

Thank you

1. On p.6 'as we do know the exact number of alleles' should, I think, read 'as we do **not** know the exact number of alleles'.

Thank you. It has now been corrected on Page 6:

“Additionally, as we do not know the exact number of alleles that contain the SNV, we iterate through all possible situations and choose the one that results in the highest likelihood L_{jk} .”

2. The meaning of the phrase 'if there are no sufficient SNVs on the chromosomes without CNA (i.e. , <100 SNVs, the default)' is unclear. My understanding of the paragraph containing this phrase is that it refers to the order of acquisition of individual SNVs. Assigning mutations collectively to a particular timepoint (before or after a copy number gain) would be a mistake, since mutations are likely to occur throughout the evolution of a tumor, both before and after copy number changes.

We absolutely agree that the reviewer mutations should not be assigned to a particular time point relevant to CNV. This sentence was referring to the case where there is no CNV (and thus no relationship with CNV needs to be considered) and we have clarified on:

“During the estimation of the number of clones, if there are enough SNVs on the chromosome sections without CNV (>100), we use these SNVs to estimate the number of clones. Otherwise all scenarios get votes in the estimation of the number of subclones, because we do not know the exact ordering of SNV and CNV events.”

Reviewer #2 (Remarks to the Author):

The authors have done a great job addressing all my previous concerns.

Thank you.